# Analysis of Video Feature Learning in Two-Stream CNNs on the Example of Zebrafish Swim Bout Classification

**Bennet Breier & Arno Onken**
School of Informatics
The University of Edinburgh
Edinburgh, EH8 9AB, UK
`b.breier@sms.ed.ac.uk, aonken@inf.ed.ac.uk`

## Abstract

Semmelhack et al. (2014) have achieved high classification accuracy in distinguishing swim bouts of zebrafish using a Support Vector Machine (SVM). Convolutional Neural Networks (CNNs) have reached superior performance in various image recognition tasks over SVMs, but their learnt features are not immediately visible. Reaching better transparency helps to build trust in their classifications and makes learned features interpretable to experts. Using a recently developed technique called Deep Taylor Decomposition, we generated heatmaps to highlight input regions of high relevance for predictions. We find that our CNN makes predictions by analyzing the steadiness of the tail's trunk, which markedly differs from the manually extracted features used by Semmelhack et al. (2014). We further uncovered that the network paid attention to experimental artifacts. Removing these artifacts ensured the validity of predictions. After correction, our best CNN beats the SVM by 6.12%, achieving a classification accuracy of 96.32%. Our work thus demonstrates the utility of AI explainability for CNNs.

## 1 Introduction

In the study by Semmelhack et al. (2014), a well-performing classifier allowed to correlate neural interventions with behavioral changes. Support Vector Machines (SVMs) were commonly applied to such classification tasks, relying on feature engineering by domain experts. In recent years, Convolutional Neural Networks (CNNs) have proven to reach high accuracies in classification tasks on images and videos reducing the need for manual feature engineering. After Lecun & Bengio (1995) introduced them in the 90s, CNNs had their break-through in the competition ILSVRC2012 with the architecture of Krizhevsky et al. (2012). Since then, more and more sophisticated architectures have been designed enabling them to identify increasingly abstract features. This development has become possible due to the availability of larger training sets, computing resources, GPU training implementations, and better regularization techniques, such as Dropout (Hinton et al. (2012); Zeiler & Fergus (2014)).

While these more complex deep neural network architectures achieved better results, they also kept their learnt features hidden if not further analyzed. This caused CNNs to come with significant drawbacks: a lack of trust in their classifications, missing interpretability of learned features in the application domain, and the absence of hints as to what data could enhance performance (Molnar (2019)). Explaining the decisions made by CNNs might even become a legal requirement in certain applications (Alber et al. (2018)).

In order to overcome these drawbacks, subsequent research has developed approaches to shed light on the inner workings of CNNs. These approaches have been successfully used for uncovering how CNNs might learn unintended spurious correlations, termed "Clever Hans" predictions (Lapuschkin et al. (2019)). Such predictions could even become harmful if the predictions entailed decisions with severe consequences (Leslie (2019)). Also, since deep neural networks have become a popular

machine learning technique in applied domains, spurious correlations would undermine scientific discoveries.

This paper focuses on zebrafish research as an applied domain of AI explainability, considering that the research community around this organism has grown immensely. The zebrafish is an excellent model organism for vertebrates, including humans, due to the following four reasons: The genetic codes of humans and zebrafish are about 70% orthologue (Howe et al. (2013)). The fish are translucent which allows non-invasive observation of changes in the organism (Bianco et al. (2011)). Furthermore, zebrafish are relatively cheap to maintain, produce plenty of offspring, and develop rapidly. Finally, they are capable of recovering their brain structures within days after brain injury (Kishimoto et al. (2011); Kizil et al. (2012)).

In this paper, we adapt CNNs to work on highly controlled zebrafish video recordings and show the utility of a recently developed AI explainability technique on this task. We train the network on optical flow for binary classifying swim bouts and achieve superior performance when compared to the current state-of-the-art in bout classification (Semmelhack et al. (2014)). We then create heatmaps over the videos with the "iNNvestigate" toolbox (Alber et al. (2018)) which highlight the areas that our CNN pays attention to when making a prediction. The resulting heatmaps show that our CNN learns reasonable features which are very different from those manually composed by Semmelhack et al. (2014).

## 2    RELATED WORK

In the following, we will give an overview over relevant CNN architectures and approaches. Then, we will summarize existing AI explainability approaches focusing on attribution techniques. Finally, we highlight important studies of behavioral zebrafish research and give details of the study by Semmelhack et al. (2014).

**CNN architectures.** Carreira & Zisserman (2017) identified five relevant types of video architectures: CNN + LSTM (Long-Short-Term Memory), 3D-CNN (Ji et al. (2013)), Two-Stream (Simonyan & Zisserman (2014)), 3D-Fused Two-Stream, and Two-Stream 3D-CNN. They differ in whether the convolutions are based on 2D or 3D kernels, whether optical flow is added, and how consecutive frames exchange information. Optical flow can be described as the horizontal and vertical displacement of a pixel from one frame to the next (Farnebäck (2003)). Several algorithms for flow calculation exist, such as TV-L1 (Zach et al. (2007)), Brox (Brox et al. (2004)), and Farneback (Farnebäck (2003)). Even novel deep learning approaches have been developed, e.g. FlowNet (Dosovitskiy et al. (2015); Ilg et al. (2017)).

Carreira & Zisserman (2017) initialized several CNN architectures with pre-trained weights and trained them on human action recognition datasets to show the benefits of transfer learning in video classifiers. Previous studies had shown this for images (Shin et al. (2016)). Also, they found that adding a temporal stream based on optical flow always improved performance. This idea of a network with a spatial stream for appearance and a temporal stream for motion had been first developed by Simonyan & Zisserman (2014).

**AI explainability techniques.** Current AI explainability techniques on images can be largely categorized into two types: attribution and feature visualization (Olah et al. (2017)). Attribution relates regions of a sample image to activations of units in the CNN, while feature visualization uncovers what kinds of inputs strongly activate a particular unit.

One of the earlier and quite successful attribution approaches is called Sensitivity Analysis (Simonyan et al. (2014)). It is based on the idea that if a single pixel were marginally changed, the prediction would worsen significantly for an important pixel, but only slightly for less important ones. Furthermore, Zeiler & Fergus (2014) and Ribeiro et al. (2016) showed simple ways of producing approximate relevance heatmaps. By occluding parts of an image one by one and observing the change in activation, they could measure the influence of different image regions. This allowed them to check whether the CNN focused on the important objects of the image or performed its classification only based on contextual information.

The technique applied in this paper is called Deep Taylor Decomposition (Montavon et al. (2017)), which arose from Layer-Wise Relevance Propagation (Bach et al. (2015)). It has been put into use

with text (Arras et al. (2017)), speech (Becker et al. (2018)), and only once with video data (Anders et al. (2018)). It highlights the areas in a sample image which the CNN deems most relevant for making a correct classification. We assume that the relevance of a pixel is determined by how much the classification accuracy would deteriorate if we altered this pixel. DTD distributes relevance from the output layer to the input layer by applying specific propagation rules for each layer. This approach equals a first-order Taylor decomposition of the output function. The authors argue that it yields a better approximation of relevance than Sensitivity Analysis. Alber et al. (2018) included this technique in their "iNNvestigate" toolbox, which we used in our work.

Apart from attribution techniques, CNNs can be explained using direct feature visualization approaches. They try to find intuitive representations of the patterns a given unit responds to particularly strongly or weakly. Among these are deconvolution networks (Zeiler & Fergus (2014)), which are closely related to Sensitivity Analysis, and optimization approaches in the input domain (Erhan et al. (2009); Olah et al. (2018)). Also, instead of creating an abstract representation of features, one can select specific samples which highly activate or suppress a particular unit. Bau et al. (2017) even went a step further by hiding irrelevant parts of the input image, similar to LIME (Ribeiro et al. (2016)).

**Behavioral zebrafish research.** As we argue in Section 1, zebrafish are a highly suitable model organism for the human body. They serve as the object of study in many fields, such as wound repair (Kishimoto et al. (2011); Kizil et al. (2012)), visual processing in the brain (Roeser & Baier (2003); Gahtan (2005); Semmelhack et al. (2014); Temizer et al. (2015)), cancer research (White et al. (2013)), and genetic modifications (Hwang et al. (2013)). Especially in neuroscientific research, understanding behavior and behavioral changes in response to cerebral interventions is of high importance (Krakauer et al. (2017)).

Previous studies therefore closely investigated the motion patterns of zebrafish (Borla et al. (2002); McElligott & O'Malley (2005)). Borla et al. (2002) described characteristic motion patterns during prey bouts, such as the precise bending of the very tip of the tail to bring the head into position and the subsequent strong arching of the tail's center. Based on their observations, they concluded that zebrafish must possess fine axial motor control in all parts of their tail.

Given that prey movements can be clearly distinguished from other types of movements, Semmelhack et al. (2014) hypothesized that there must be a dedicated circuitry in the zebrafish brain. They found a pathway from retinal ganglion cells to an area called AF7 projecting to the optic tectum, the nucleus of the medial longitudinal fasciculus, and the hindbrain, which in turn produces the characteristic motor output. They verified their findings by ablating the AF7 neuropil and observing that lesioned fish failed to respond to prey stimuli with a movement that a trained SVM would classify as prey. They have identified the following five features as the most discriminating ones, ordered by descending importance:

1. Maximum tail curvature (maximum over the bout)
2. Number of peaks in tail angle
3. Mean tip angle (absolute value of tip angle in each frame, average over the bout)
4. Maximum tail angle (maximum over the bout)
5. Mean tip position (average position of last eight points in tail, with horizontal deflection as a fraction of the tail length)

In our work, we make use of the dataset they gathered for training their SVM and compare our CNN to the features above.

## 3 METHODS

We trained a Two-Stream CNN to distinguish prey and spontaneous swim bouts of larval zebrafish with short video samples. For data preparation, we first extracted standardized snippets from raw videos and then performed augmentation by subsampling, flipping, and cropping. After training, we computed heatmaps showing the regions of high relevance within frames[1].

---

[1]The code can be found at `https://github.com/Benji4/zebrafish-learning.git`

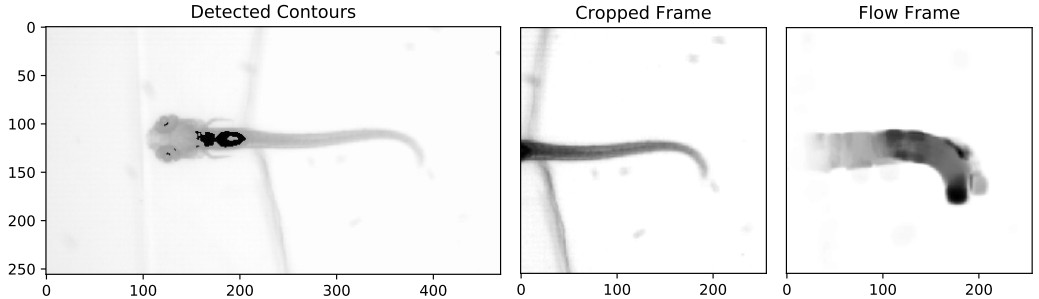

Figure 1: Detected contours (black) in the original frame, resulting cropped frame after normalization, and corresponding optical flow frame.

**Data pre-processing.** We used the raw video files recorded by Semmelhack et al. (2014) with a high-speed camera at 300 frames per second and labeled as either spontaneous (0/negative, 56.1%) or prey (1/positive, 43.9%) bout. The heads of the fish were embedded in a substance called agarose to keep them steady. We turned the videos into grayscale, normalized them, and kept a crop of size 256x256 pixels from each frame, such that the right-most part of the bladder was central on the left, as shown in Figure 1. We did not include the head of the fish in the crop, because the eyes would give away information about the type of bout (Bianco et al. (2011)). More details in Appendix B.

For correct centering we implemented a gamma correction with $\gamma = \exp\left(-\text{skewness}/\text{param}\right)$, (where param was tweaked to maximize detection performance (here param = 4.3)) and applied a binary threshold at value 3 to separate the bladder from the rest of the fish. Since the eyes might also fall under this threshold, we declared the right-most contour the bladder. Each raw video contained several bout events which we extracted as sequences of 150 frames resulting in 1,214 video snippets. Each sequence started 15 frames before the actual bout motion was detected.

**Extension to the pre-processing procedure.** After training and heatmap analysis, we found that the trained CNN had found a shortcut for classification by using agarose motion instead of tail features. We therefore extended our pre-processing procedure by setting squares of 85x85 pixels in the upper and lower left-hand corners of all frames plain white. While this cut away some of the tail tip in rare occasions, it ensured that bouts could not be classified based on the agarose motion anymore.

**Data augmentation.** Our data augmentation procedure worked with batches of 32 videos with the original data randomly shuffled, allowing a decent approximation of the gradient during training with some inherent noise, which is generally desirable to avoid falling into sharp minima (Keskar et al. (2016)). The procedure performed three augmentation steps: first subsampling, then flipping, and cropping, which achieved augmentation factors of 8, 2, and 9 respectively, totaling 174,816 augmented samples. We decided to compute optical flow after subsampling, not before, in order to create differing augmented samples. All samples of an augmented batch were derived from different original videos to ensure a good gradient approximation during training.

In the subsampling step, our algorithm randomly selected 86 out of 150 frames under the constraint that between two frames, no more than 2 frames may be omitted. This was to ensure meaningful flow calculation in the subsequent step, because the tail could move quite fast. After subsampling, for each video our procedure selected one of the 86 frames to be the input for the spatial network. Then we computed the optical flow resulting in 85 flow frames with one $x$ and $y$ component each. They were stacked to 170 channels, alternating $x$ and $y$. We used the optical flow algorithm by Farnebäck (2003) with parameters[2] detecting flow even when the tail moved fast. The procedure then generated 18 augmented batches from each subsampled batch by randomly flipping vertically

---

[2]pyramid scale = 0.8, pyramid levels = 10, window size = 10, iterations per pyramid level = 10, pixel neighborhood for polynomial expansion = 13, std dev of the Gaussian for polynomial expansion = 1.8.

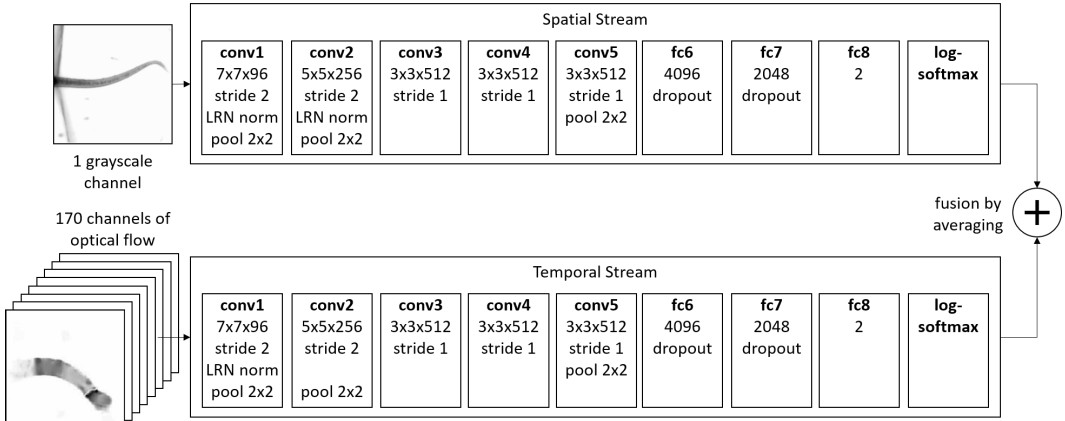

Figure 2: Full Two-Stream CNN architecture.

and cropping into crops of size 224x224 with the upper left corner at coordinates (8,8), (8,16), (8,24), (16,8), (16,16), (16,24), (24,8), (24,16), and (24,24).

Since flow was calculated as floats, we minimized storage space significantly by rescaling each frame to range 0–255, compressing them with lossy JPEG-compression at level 40, and turning them into unsigned integers, similar to what Simonyan & Zisserman (2014) had done. Frames were rescaled later for training.

**CNN architecture and framework.** Just like Simonyan & Zisserman (2014), we used a two-stream network with an adapted CNN-M-2048 network (Chatfield et al. (2014)) for each stream, as depicted in Figure 2. As shown by Carreira & Zisserman (2017), this network can deal with a small sample size and learns quickly. The spatial stream had one gray-scale channel and the temporal stream 170 flow channels in the first layer. After obtaining the predicted probabilities of each stream by calculating the log-softmax of the individual two outputs, they were fused by averaging. We computed the negative log-likelihood loss of this average, which can be interpreted as the joint log-probability of both streams, assuming their statistical independence.

Our dataset was made up of 38 files, with 28, 4, and 6 files for training, validation, and test sets respectively. The individual sets were not shuffled during training in order to allow sequential reads, which might have decreased training time. Notwithstanding, batches consisted of random samples due to our augmentation procedure explained above.

**Initialization of weights.** We initialized both streams with weights pre-trained on ImageNet[3]. This has become a common initialization strategy in various classification tasks (Shin et al. (2016); Lee et al. (2016); Van Horn & Perona (2017)), due to the utility of general features learnt from natural images, such as edge detection. While Simonyan & Zisserman (2014) did not pre-train flow, Carreira & Zisserman (2017) found a pre-trained temporal stream to reach superior performance. With this initialization we hoped that training would require only fine-tuning and therefore less data and fewer epochs.

Specifically, for the weights of the input layer of the spatial stream we took the average of the pre-trained weights of 3 RGB-channels to get the weights for 1 grayscale-channel. For the temporal stream we copied the RGB-channels $56\frac{2}{3}$ times to get 170 channels, and added uniform random noise to all of them. This was to ensure that the channels evolved differently during training and should aid learning. Regarding outputs, we averaged the pre-trained weights of 500 units on the output layer to obtain the weights of two output neurons, because we dealt with 2 classes instead of 1,000 as in ImageNet.

**Training procedure.** We made use of the Adam optimizer with standard settings and tuned its learning rate and weight decay coefficient – the neural network equivalent of L2-regularization (Loshchilov & Hutter (2019)). Furthermore, we added a gamma learning rate scheduler which

---

[3]http://www.vlfeat.org/matconvnet/models/imagenet-vgg-m-2048.mat

Table 1: Results on training, validation, and test (split by stream) sets for baseline (B), analyzed CNN (0), best CNN before (1) and after (2) correction.

| ID | LR | WD | TRAIN | VALID | SPATIAL | TEMPORAL | FULL TEST |
|----|------|------|-------|-------|---------|----------|-----------|
| B | – | – | .946 | .946 | – | – | .902 |
| 0 | 1e-4 | 1e-3 | .9962 | .9489 | .8216 | .9441 | .9596 |
| 1 | 1e-5 | 1e-3 | .9869 | .9372 | .8138 | .9660 | .9669 |
| 2 | 1e-5 | 1e-3 | .9997 | .9597 | .8213 | .9581 | .9632 |

updated the learning rate by a multiplicative factor of $\gamma = 1/\sqrt{epoch}$ (Kingma & Ba (2014)) every epoch. Our training framework computed accuracy on the validation set after each epoch to estimate generalization performance.

Since we were initializing our CNN with weights learned from quite a different domain, fine-tuning was crucial. In particular, we performed a hyperparameter search over learning rate ({1e-3, 1e-4, 1e-5}) and weight decay ({1e-2, 1e-3, 1e-4}) using a smaller dataset and training for about 8 epochs. With the best hyperparameters we trained the CNN on the full dataset for 5 epochs.

**Relevance analysis with heatmaps.** Making our CNN more transparent required an AI explainability technique which would be well interpretable in the optical flow domain of our temporal stream. We expected feature visualization approaches (Olah et al. (2018); Bau et al. (2017); Erhan et al. (2009)) and deconvolutions (Zeiler & Fergus (2014)) to yield less interpretable outputs than attribution techniques. In particular, we chose to analyze our CNN with Deep Taylor Decomposition (DTD), because it had been applied successfully before (Lapuschkin et al. (2019); Van Molle et al. (2018); Anders et al. (2018)) and was conveniently accessible in the "iNNvestigate" toolbox (Alber et al. (2018)). For validation purposes, we additionally generated saliency maps (Simonyan et al. (2014)) and heatmaps from Guided BackProp (Springenberg et al. (2015)). The produced relevance heatmaps could be expected to give clues about what specific regions of optical flow, within a frame and across frames, the network was paying attention to.

Also, we simplified the analysis by splitting the network into its individual streams. This was possible because no weights were learned after the final layer of each stream. Once the network was initialized correctly, "iNNvestigate" made the generation of heatmaps surprisingly simple. Also, with about 50 minutes for the whole analysis it was quite fast even on CPU, because it effectively only needed one forward and backward pass per sample. We used a small dataset of 3,420 samples for analysis by setting the subsampling factor from 8 to 1, in order to simplify the process.

## 4 RESULTS

We fine-tuned the training of our CNN to reach high accuracies in distinguishing prey bouts of larval zebrafish from spontaneous swims. Subsequently, we analyzed the learned weights by computing and averaging the relevance heatmaps of all samples grouped by class. We further define prey bouts as positive and spontaneous swims as negative.

**CNN test results.** We performed a small hyperparameter search over learning rate and weight decay, which proved sufficient because all models were initialized with pre-trained weights from ImageNet. For our baseline SVM – detailed in Appendix A – we report the 5-fold cross-validated accuracy and the final accuracy on the held-out test set. The hyperparameters[4] agree with the ones found by Semmelhack et al. (2014). We further present the accuracies of the CNN used for heatmap analysis, as well as the best CNNs before and after removal of experimental artifacts on training, validation, and test sets in Table 1. For the test set we additionally report individual accuracies of the spatial and temporal streams. We highlight that the final CNN attains a test accuracy of 96.32%, which is 6.12% points better than the baseline.

**Relevance heatmaps.** We used relevance heatmaps to visualize the regions the CNN pays most attention to when classifying a specific sample. We computed relevance averages across samples and

---

[4]RBF-kernel with $\gamma = 0.001$ and C = 1

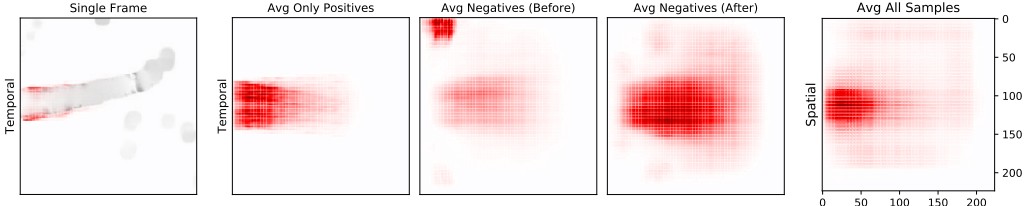

Figure 3: Relevance heatmaps of the CNN. High relevance is shown in dark red color, while light red stands for low relevance. Left: example of a temporal heatmap over a single frame (true positive, optical flow in grayscale, relevance heatmap overlaid in red). Middle: averages of temporal heatmaps split by class; negatives before and after correction. Right: average of spatial heatmaps over all samples.

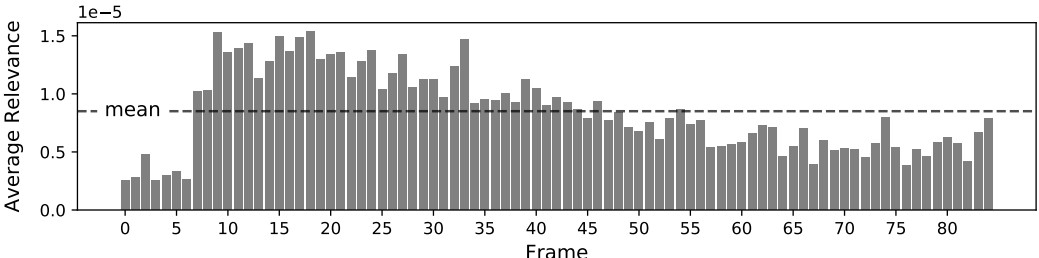

Figure 4: Distribution of relevance on the flow frames of the averaged sample.

frames, as well as split by class in Figure 3 for more comprehensive insights into the features learned by our CNN. Similar results from other explainability techniques can be found in Appendix C. As expected, the heatmaps exhibit the checkerboard artifacts typical of kernels with stride 2 in the first convolutional layer (Odena et al. (2016); Montavon et al. (2017)).

**Steadiness of the fish's trunk as differentiating feature.** First and foremost, the heatmaps show that our CNN is able to differentiate the movements of zebrafish based on their characteristic motion. Relevance is highly concentrated at the trunk, i.e. the rostral part of the tail, for both temporal and spatial stream. We observe a very sharp relevance pattern along the edges of the tail. This indicates that the pre-trained weights helped the network to look for edges. The CNN pays little to no attention to the end of the tail even though it is visible in most frames. Instead, it makes positive classifications by looking at the very start of the trunk. The heatmaps suggest that a calm and steady trunk indicates a prey bout. As for the negatives, the spread out relevance pattern reflects the high frequency of tail deflections typical of spontaneous bouts, which had been identified by Semmelhack et al. (2014) before. This makes clear that the network is able to differentiate the movements of zebrafish to high accuracy based on their characteristic motion.

**"Clever Hans" predictions.** CNNs are incredibly powerful at finding any kinds of correlations in the input data even if they are not related to the object of interest. Lapuschkin et al. (2019) have termed such spurious correlations "Clever Hans" predictions, because the model bases its prediction not on what we want it to focus on, but some unintended artifacts in the data. Figure 3 shows clearly that our CNN bases a significant number of its negative responses mainly on motion in the top left corner and focuses little on tail appearance and motion. This plays a role only in negative classifications and only in the temporal stream. While the heatmaps are vertically symmetric, as we would expect due to vertical flipping during augmentation, this is not true for the peculiar region in the top left corner. Figure 3 depicts the averaged heatmap after removing the artifacts in the top and bottom left hand corners and retraining our CNN. Relevance is now entirely focused on the tail.

**Relevance distribution across frames.** Most relevance is concentrated on the frames in the range 7–46, as depicted in Figure 4. The first seven frames are of least importance. This is very likely because our pre-processing procedure added a buffer of 15 frames before each bout, suggesting that

the network focuses on the range of frames which are in fact the most relevant ones. This further supports the hypothesis that our CNN is able to differentiate zebrafish movements based on their characteristic motion patterns.

## 5   CONCLUSION AND FUTURE WORK

We trained a two-stream Convolutional Neural Network (CNN) on recordings of larval zebrafish to classify prey and spontaneous swim bouts. We then visualized the learned weights by generating relevance heatmaps showing which regions of the input the network focuses on while performing its classifications. We find that our CNN is capable of learning highly discriminating tail features. These features seem to be quite different from the ones used in the SVM classification by Semmelhack et al. (2014) - the previous state-of-the-art in bout classification. The heatmaps further uncovered a "Clever Hans" type of correlation. After removing this spurious correlation and retraining the network, the network reached a test accuracy of 96.32%, which is 6.12% points better than the accuracy achieved by Semmelhack et al. (2014). Judging from the test accuracy, our CNN has learned better discriminating features than those used for the SVM by Semmelhack et al. (2014), and has thus beaten manual feature engineering in this application domain.

**Steadiness of the fish's trunk as differentiating feature.** The relevance heatmaps and high accuracy show that the network achieves correct classifications by looking for salient features in the trunk of the tail while largely disregarding the tip. A sharp and clear relevance profile confined to the edges of the trunk gives a clear sign of a prey bout. The opposite speaks for a spontaneous bout. Here, attention spreads out to capture the strong vertical oscillation of the trunk. For this reason we conclude that the CNN makes its predictions based on the steadiness of the trunk. We believe our interpretation of learned features to be in line with existing research on the kinematics of prey bouts. As shown by Borla et al. (2002) and McElligott & O'Malley (2005), prey bouts require fine control of the tail's axial kinematics to perform precise swim movements. Zebrafish noticeably reduce their yaw rotation and stabilize the positioning of their head to make a targeted move at their prey. Such precise movements are not required in spontaneous swim bouts. The heatmaps indicate that the network has found clear evidence for these kinds of motion in the trunk of the tail.

Furthermore, we argue that the CNN has learned features which are very different from the ones identified by Semmelhack et al. (2014). All of their features – as outlined in Section 2 –, except the second one, rely on information from the tip of the tail and a complete sequence of frames. However, many optical flow frames do not depict the tip of the tail because of its small size and high speed. This might have happened due to suboptimal parameter settings which could not handle the sometimes long distances which the tip traveled between frames. Also, subsamples include only 85 of the original 150 frames for each video. Due to its higher performance, we conclude not only that the CNN has learned a different set of features, but also that these features must bear higher discriminative power.

**Origin of the "Clever Hans" correlation.** The telltale motion in the top left corner stems from a substance called agarose, which the fish's head was embedded in to keep it steady. It is quite curious that, while not visible to human eyes, the agarose seems to be moving each time the fish performed a spontaneous swim bout, but not so for a prey bout. We speculate that this correlation was unintentionally introduced by the experimenters who might have tapped the petri dish to induce the fish to perform a spontaneous swim bout.

**Future work.** Calculating and storing optical flow is expensive. If we attained similar performance on original frames, training would be considerably cheaper. While we can confirm the findings by Simonyan & Zisserman (2014) that the spatial stream by itself reaches a fairly competitive accuracy, it provides only very minor improvement to the overall network. Yet, this stream is probably looking for very similar features as the temporal stream, because it focuses largely on the upper half of the tail, just like the temporal stream. If that is the case, we should see improved performance when giving the spatial stream a sequence of frames. It should be interesting to probe whether the spatial stream could then match or even surpass the performance of the temporal stream.

Furthermore, CNNs such as the one used in this paper could be used to investigate brain recovery in larval zebrafish. It has been shown on a cellular level that zebrafish can heal their brain within days after a lesion. However, this needs to be proven on a behavioral level (Krakauer et al. (2017)).

Future work could perform a lesion study on the optic tectum in zebrafish (McDowell et al. (2004); Roeser & Baier (2003)), a brain region responsible for translating visual input into motor output. CNNs could then assess swim bouts of recovered fish and give a measure for potential behavioral changes. Insights from relevance heatmaps would be required if the CNN were not able not distinguish recovered fish from healthy ones.

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

## A  FITTING THE BASELINE SVM

For each frame in the 1,214 videos, we applied the tail-fitting code developed by Semmelhack et al. (2014) to compute points along the tail, as depicted in Figure S5. We initialized their procedure central and 8 pixels from the left edge, because after pre-processing we could assume this to be just next to the right end of the bladder. Some of the videos contained frames which the tail-fitting code had problems processing, possibly because the pre-processing procedure cut off the tip of the tail in these instances. This resulted in 953 correctly processed videos, including 482 (50.6%) spontaneous and 471 (49.4%) prey bouts. We performed no augmentation here because this would not have benefited the SVM.

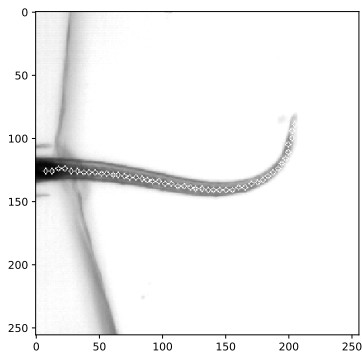

Figure S5: Distribution of points along the tail after running the tail-fitting procedure.

The feature extraction and model fitting algorithm then split the set into 85% training and 15% held-out test sets. Semmelhack et al. (2014) have identified 5 key features which allowed their SVM to achieve a cross-validation accuracy of 96%. They did not report results on a held-out test set. We used their provided code to extract these features. Then we performed a grid-search to tune SVM-kernel, $\gamma$, and C[5]. Just like Semmelhack et al. (2014) we used stratified 5-fold cross-validation (Kohavi (1995)).

## B  PROJECT PIPELINE

Figure S6 gives an overview over the whole project pipeline. The figure includes a depiction of the raw data input, pre-processing, augmentation, CNN and SVM training, and heatmap analysis. Figure S7 summarizes the data augmentation procedure. All scripts worked with a seed of 462019. We used the openly available distributions of NumPy 1.16.4 (van der Walt et al. (2011)),

---

[5]RBF-kernel, $\gamma \in \{$1e-1, 1e-2, 1e-3, 1e-4$\}$, C $\in \{$0.01, 0.1, 1, 10$\}$; linear-kernel, C $\in \{$0.01, 0.1, 1, 10$\}$.

Matplotlib 3.1.1 (Hunter (2007)), tqdm 4.32.2 (da Costa-Luis (2019)), OpenCV 4.1.0.25 (Bradski (2000)), scikit-learn 0.21.2 (Pedregosa et al. (2011)), PyTorch 1.1.0 (Paszke et al. (2017)), h5py 2.9.0 (Collette (2013)), TensorFlow 1.14.0 (Abadi et al. (2015)), Keras 2.2.4 (Chollet et al. (2015)), and iNNvestigate 1.0.8 (Alber et al. (2018)).

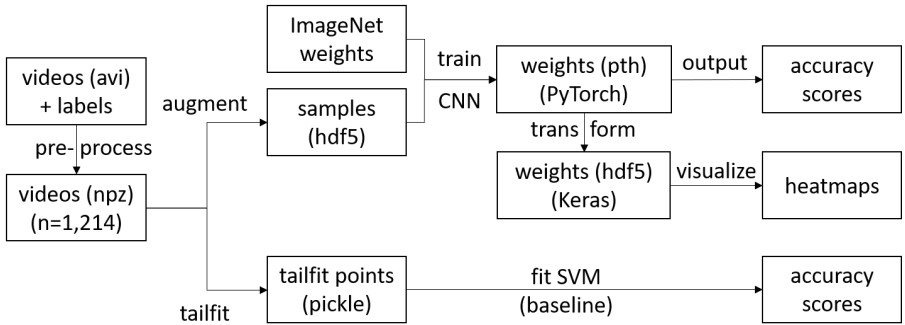

Figure S6: Overview over the whole project pipeline.

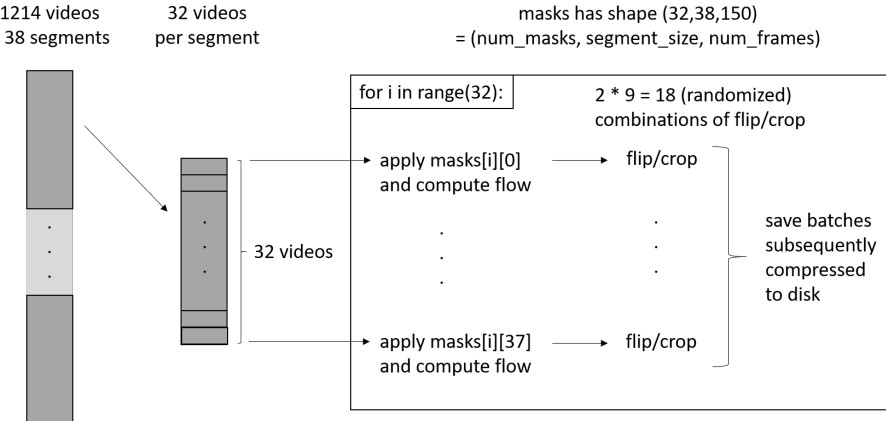

Figure S7: Data augmentation procedure.

**Pre-processing.** We aimed to center the fish's bladder on the left of each cropped frame. To achieve this, we applied a binary threshold at value 3, because after normalization and gamma correction the pixels of the bladder were separated mostly below this value. While the fish was quite light, the eyes as the second darkest part of the fish might still have fallen under this threshold. Therefore, we first detected all contours to discard tiny contours ($< 0.01\%$ of the whole frame) and then kept only the right-most contour. Since this had to be the bladder now, we could get the crop dimensions using the right-most pixel of that contour.

Each raw video mainly consisted of a still fish interspersed with a few short bouts. These were the events we extracted into consecutive 150 frames each. The idea was to detect motion by checking the percentage of pixel value changes from one frame to the next, considering only the tail. We omitted pixels other than the tail with a simple binary threshold at value 200. The pixels had to change in our case at least 0.38% of the entire pixel range (height $\times$ width $\times$ 255) in order for motion to be detected. If the algorithm detected motion for a certain number of consecutive frames, it set this as the start of an event. Also, it added a preceding buffer of 15 frames. The end was set 150 frames from the start. If no motion was detected, we automatically took the first frame as a start.

We had to take care that extracted videos did not overlap, i.e. in part contained identical frames, even if there were two or more distinct movements contained within 150 frames. This might have otherwise lead to train/test-contamination. Therefore, we discarded any detected motions which fell

in the range of a previous video. One special case we did not take into account was when the start was less than 150 frames from the end of the file. We shifted the start back by just enough frames to fit it in, but this might have made it overlap with a previous video. Since this case was probably very rare and not detrimental, we have kept the code as it was.

**Data augmentation.** We parallelized data augmentation with 38 workers, because optical flow calculation took roughly 14 seconds per subsample. We used a Dell PowerEdge R815 with four 16 core Opteron CPUs, 256 GB memory, and 4 TB of disk space (HDD). Furthermore, the resulting hdf5-files were compressed with gzip-compression at maximum compression level. This outperformed lzf-copmression by a factor of 1.76 when comparing training times. This advantage can be ascribed to heavy parallelization of decompression and relatively low transfer speeds between hard drive and memory.

**Training procedure.** We implemented PyTorch's `Dataset` module to make use of the multi-processing capabilities of the `DataLoader` class on servers with 16–32 CPU cores and 4 GPUs (NVIDIA GTX1060 6 GB) each. This was necessary to achieve manageable epoch times.

**Relevance analysis with heatmaps.** The toolbox "iNNvestigate" (Montavon et al. (2017)) for analyzing the trained weights only supported Keras with the TensorFlow-backend. Therefore, we re-implemented the exact structure of our CNN and initialized it with the extracted weights from PyTorch. While the conversion could have been done with tools like ONNX, after a few unsuccessful attempts we transported the weights with a custom Python script.

A caveat to the "iNNvestigate" toolbox emerged after heatmap generation: it had problems analyzing 1,578 of the 3,420 samples. These produced an empty output. We made sure the problematic samples did not follow any systematic pattern by checking the indices of correctly analyzed samples, the ratio of true positives and negatives and false positives and negatives, as well as the distribution of classification confidence after the analysis. Since all numbers were the same as before the analysis, we continued with 1,842 samples for further investigation.

## C  OTHER EXPLAINABILITY TECHNIQUES

We generated saliency maps (Simonyan et al. (2014)) and Guided BackProp heatmaps (Springenberg et al. (2015)) analogous to the relevance heatmaps in Figures S8 and S9 for comparison with the more recent technique of DTD. It becomes apparent that these other two techniques allow similar insights, although slightly fuzzier. Importantly, they also uncover the "Clever Hans" prediction.

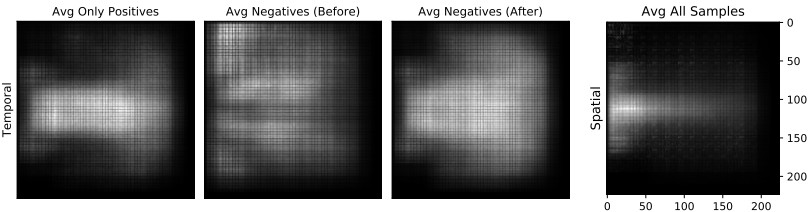

Figure S8: Averaged saliency maps, analogous to Figure 3.

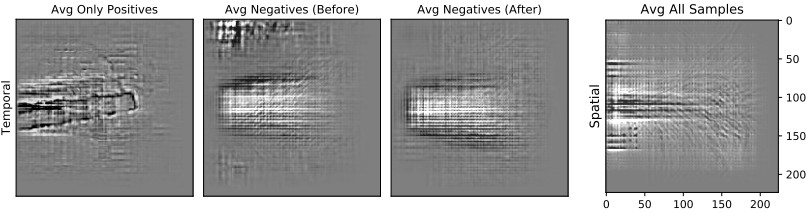

Figure S9: Averaged heatmaps from Guided BackProp, analogous to Figure 3.

## D    SELECTION OF ADDITIONAL HEATMAPS

Here, we depict the ten most informative consecutive flow frames of the single most confident true positive (Figure S10), true negative (Figure S11), false positive (Figure S12), and false negative (Figure S13) sample. Figure S15 summarizes the spatial heatmaps of the same samples. Moreover, we gather five particularly informative flow frames in Figure S14 and four spatial heatmaps in Figure S15.

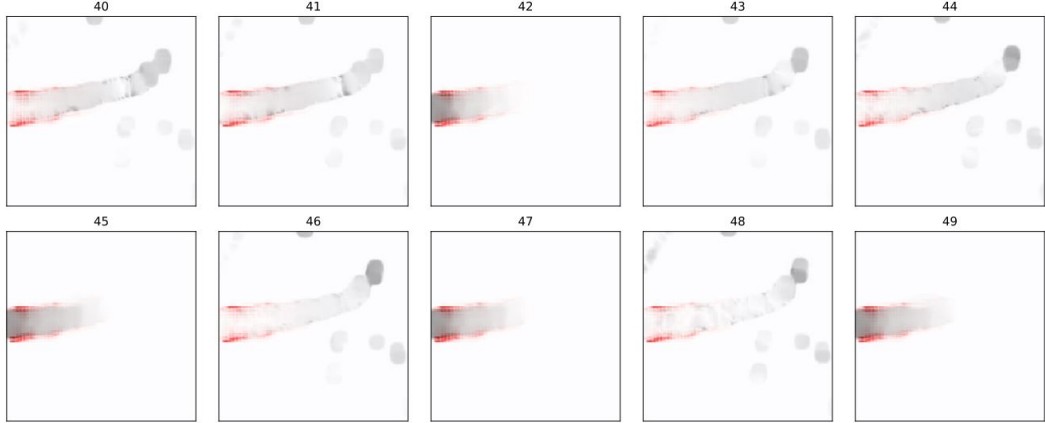

Figure S10: Selection of flow frames of the most confident true positive sample.

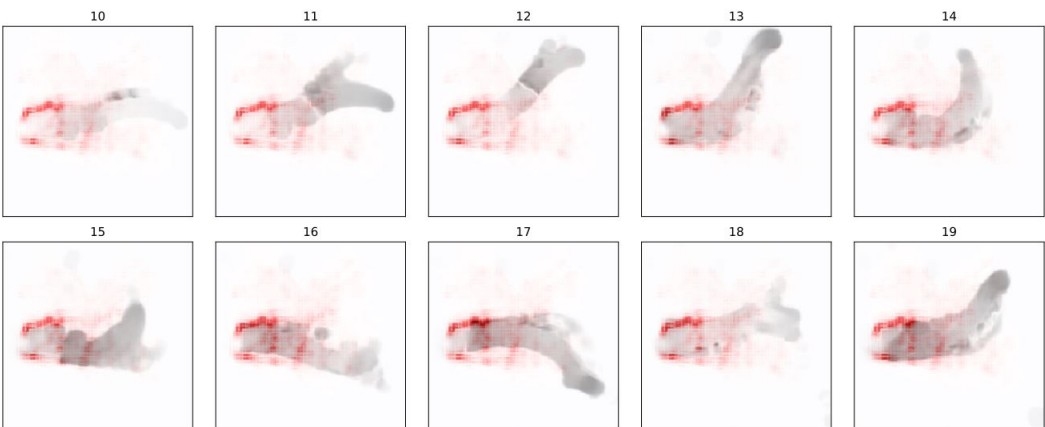

Figure S11: Selection of flow frames of the most confident true negative sample.

## E    CONFIDENCE ANALYSIS ON HEATMAPS

We performed a confidence analysis on the heatmaps which depicted the "Clever Hans" feature of agarose motion in the top left corner. We sorted all negative classifications by increasing confidence calculated as $-\log\big(\log(\mathrm{P}(1))/\log(\mathrm{P}(0))\big)$ for each sample. We then grouped and averaged the heatmaps over windows of 104 samples, as shown in Figure S16. The analysis uncovered that the more confident a negative classification, the more the CNN relied on tail features. This in turn indicated that the CNN was able to learn actual features on the tail and did not entirely rely on agarose motion. Also, it suggested that tail features were a better predictor than agarose motion.

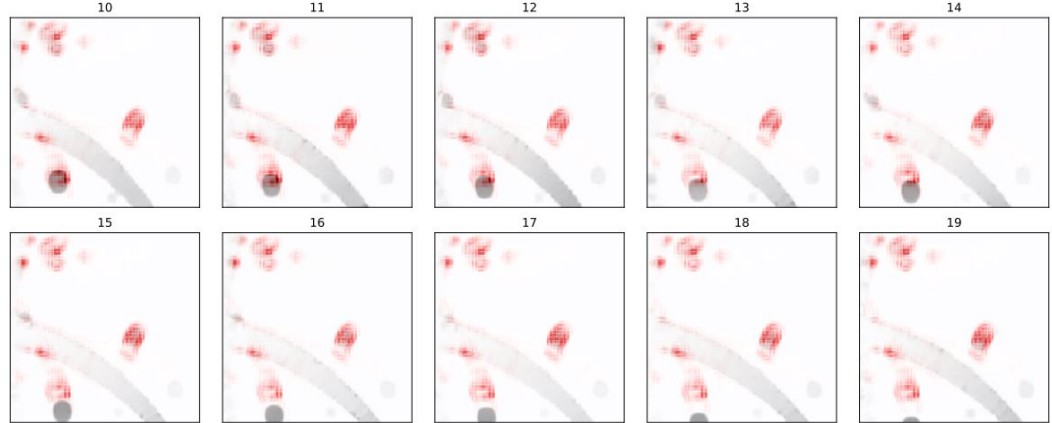

Figure S12: Selection of flow frames of the most confident false positive sample.

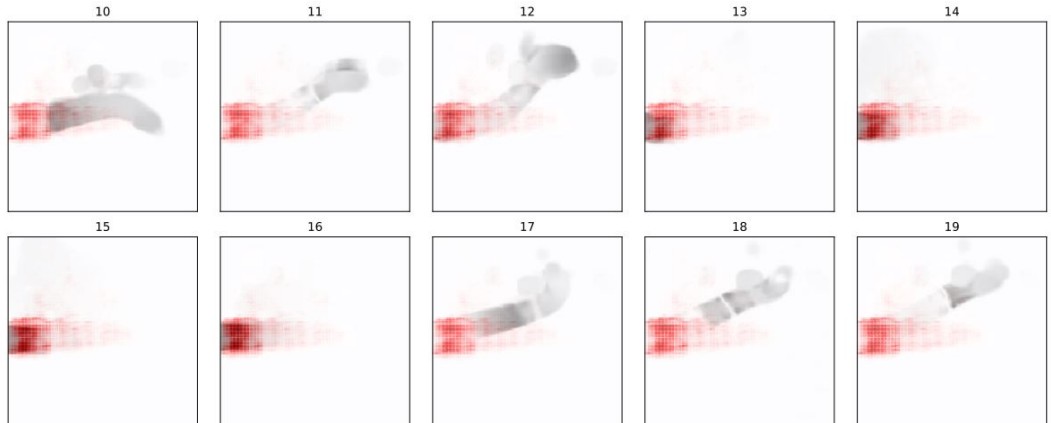

Figure S13: Selection of flow frames of the most confident false negative sample.

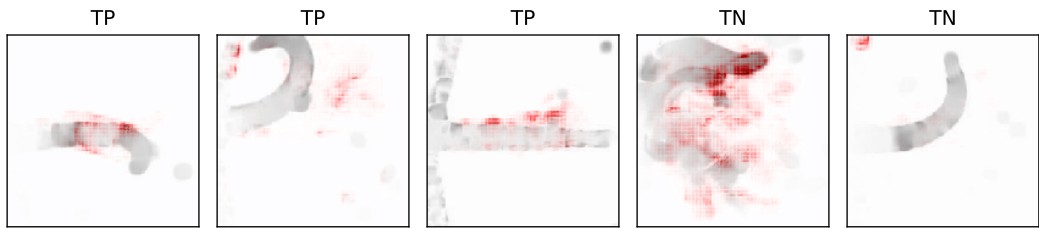

Figure S14: Selection of flow frames of a selection of true positive (TP) and true negative samples (TN).

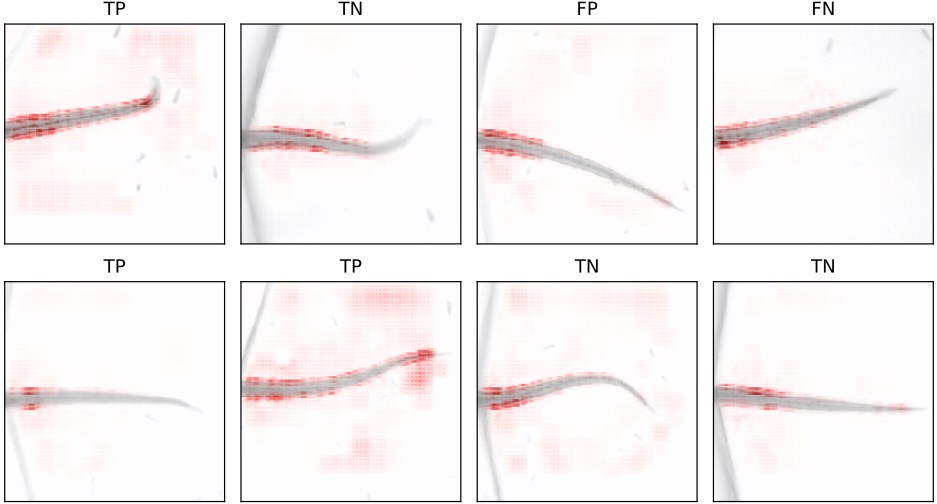

Figure S15: Upper row: spatial input of the most confident samples of each category. Lower row: spatial input of a selection of samples.

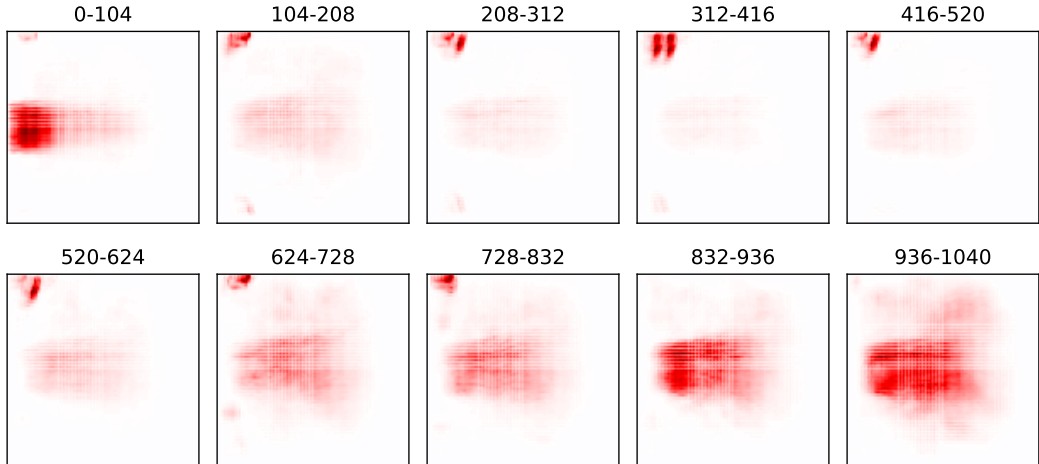

Figure S16: Averaged heatmaps of 104 samples per subfigure, sorted by increasing confidence for responding negative.

