# OpenReview forum: "Analysis of Video Feature Learning in Two-Stream CNNs on the Example of Zebrafish Swim Bout Classification"
_ICLR.cc/2020/Conference — Accept (Poster)_

### Official Review · AnonReviewer1 · 2019-10-24
**Official Blind Review #1**

**Rating:** 6

**Review:**

the paper uses model interpretation techniques to understand blackbox CNN fit of zebrafish videos. they show that, relying on the technique of deep taylor decomposition, their CNN relies its prediction on a different part of zebra fish than existing understanding. it is also able to detect the use of experimental artifacts, whose removal improves predictive performance.

the idea of a case study about the usefulness of model interpretation techniques is interesting. while the experimental studies rely on our belief that the interpretation technique indeed interprets, the result that removing experimental features and improving predictive performance is convincing and interesting. it illustrates how model interpretability and human intuition and domain knowledge can be useful.

**Experience Assessment:**

I do not know much about this area.

**Review Assessment: Checking Correctness Of Derivations And Theory:**

I assessed the sensibility of the derivations and theory.

**Review Assessment: Checking Correctness Of Experiments:**

I assessed the sensibility of the experiments.

**Review Assessment: Thoroughness In Paper Reading:**

I read the paper at least twice and used my best judgement in assessing the paper.

---

> ### Author Response · Authors · 2019-11-12
> **Reply to reviewer**
>
> Thank you for your kind review and your accurate assessment. That is exactly what we wanted to demonstrate.

---

### Official Review · AnonReviewer3 · 2019-10-31
**Official Blind Review #3**

**Rating:** 6

**Review:**

SUMMARY: explore the use of CNN in a binary task on images of zebrafish

It is important to note that researchers in the field of AI and deep learning are themselves aware of the fallacies of deep learning, and are striving everyday to overcome these themselves. The hype over deep learning has caused certain disdain among a section of the research community over the workings of deep neural networks. This is evident in this paper with the authors calling CNNs "black box" and the learnings of a neural network "cheating". Perhaps the authors are not aware that CNNs are hardly black boxes, their inner workings quite transparent in mathematical terms, which the submitted paper itself explores. Perhaps the authors are also not aware that the fallacies that causes CNNs to overfit on some characteristics in the input data are also present in other machine learning tools such as SVMs. Perhaps the intention of the authors is to bring more relevance to the dangers of spurious correlations, especially when the applications are critical. I hope the community can work together in improving the state of the art while improving transparency and explainability.

BACKGROUND:
Hypothesis: Prey movements in zebrafish are characterized by specific motions, that are triggered by a specific pathway involving an area called AF7.
Validation: Semmelhack et. al. (2014) removed the AF7 neuropil and observed that they failed to respond to prey stimuli
AI: the prey stimuli was a characteristic movement that an SVM was trained to detect.

Good to know that the authors will share their code.

It is not explained why the authors chose to pretrain on ImageNet, since ImageNet does not have any image classes that are comparable to the dataset the authors use. They then proceed to do a hyperparameter sweep to fine-tune on their dataset.

It is also not explained why they chose to average outputs of 500 output nodes to get two outputs, instead of simply replacing the last layer with a 2-neuron layer and finetune, as is general practice.

The level of detail in the training procedure is very helpful to reproduce the setting as well as establish a reference for any future work in this direction. This itself is a notable achievement and a good use of significant research time. Furthermore, the authors conduct one good analysis (DTD) to explain the results of their CNN. The conclusion has many repeated points from previous sections, but it is a good summary.

Given their premise about explainability in machine learning, perhaps more significance was to be given to DTD, and other methods also performed to check if the results coincide with those from DTD.

**Experience Assessment:**

I have read many papers in this area.

**Review Assessment: Checking Correctness Of Derivations And Theory:**

N/A

**Review Assessment: Checking Correctness Of Experiments:**

I carefully checked the experiments.

**Review Assessment: Thoroughness In Paper Reading:**

I read the paper thoroughly.

---

> ### Author Response · Authors · 2019-11-12
> **Reply to reviewer**
>
> Thank you for your detailed review and for pointing out issues that need further clarification. We hope our revisions will address your concerns.
>
> We revised problematic wording in the manuscript. Terms like "black box" and "cheating" were unnecessary exaggerations which we have now replaced with expressions such as "hidden if not further analyzed" and "found a shortcut" that better reflect our intentions.
>
> Furthermore as you suggested, we added a justification for why we used pre-trained weights from ImageNet as initialization values for our network and clarified the description of our initialization strategy:
> "We initialized both streams with weights pre-trained on ImageNet\footnote{http://www.vlfeat.org/matconvnet/models/imagenet-vgg-m-2048.mat}. This has become a common initialization strategy in various classification tasks (\cite{Shin2016, Lee2016, VanHorn2017}), due to the utility of general features learnt from natural images, such as edge detection. While \cite{Simonyan2014} did not pre-train flow, \cite{Carreira2017} found a pre-trained temporal stream to reach superior performance. With this initialization we hoped that training would require only fine-tuning and therefore less data and fewer epochs."
>
> As you correctly pointed out, it is common practice to train a 2-neuron output layer, which is exactly what we did. We did not average the outputs but the weights of the 500 output units in order to initialize the weights of 2 neurons. We clarified this in the manuscript:
> "Regarding outputs, we averaged the pre-trained weights of 500 units on the output layer to obtain the weights of two output neurons, because we dealt with 2 classes instead of 1,000 as in ImageNet."
>
> Finally, to address your concern regarding additional explainability techniques besides DTD, we have now created saliency maps as well as heatmaps from Guided BackProp. We found that these other two techniques allow similar insights, although slightly fuzzier. Importantly, these techniques also uncover the "Clever Hans" prediction. We now briefly mention this additional analysis in the main text and added a section with additional figures to the Appendix.

---

### Official Review · AnonReviewer4 · 2019-11-01
**Official Blind Review #4**

**Rating:** 6

**Review:**

The paper presents a case study of training a video classifier using convolutional networks, and on how the learned features related to previous, hand-designed ones. The particular domain considered is of importance for biologists/medical doctors/neuroscientists: zebra fish swim bout classification.

In order to identify which particular features the neural networks are paying attention to, the paper used Deep Taylor Decomposition, which allowed the authors to identify "clever-hans"-type phenomena (network attending to meaningless experimental setup differences that actually gave a way the ground truth classes). This allowed for the authors to mask out such features and make the network attend to more meaningful ones. In particular observations like " looking for salient features in the trunk of the tail while largely disregarding the tip", are typically absent from most deep learning studies and it's quite interesting.

Overall the paper is well written, the experiments are well designed; everything seems very rigorous and  well executed. It makes for a very good quality practitioner-level case study on video understanding, which may also be useful for people studying zebra fish or related simple life forms. My main concern with the paper is whether ICLR is an appropriate venue, as it does not provide pure machine learning contributions in the form of new techniques of generally applicable insights.

**Experience Assessment:**

I have published one or two papers in this area.

**Review Assessment: Checking Correctness Of Derivations And Theory:**

N/A

**Review Assessment: Checking Correctness Of Experiments:**

I assessed the sensibility of the experiments.

**Review Assessment: Thoroughness In Paper Reading:**

I read the paper at least twice and used my best judgement in assessing the paper.

---

> ### Author Response · Authors · 2019-11-12
> **Reply to reviewer**
>
> Thank you for your concise and insightful review.
>
> Your concern regarding the appropriateness of ICLR as a venue for publication is very understandable considering that the paper is built around behavioral zebrafish research. Nevertheless, our work supports the conference in exploring relevant topics around supervised representation learning, visualization and interpretation of learned representations, and applications in neuroscience. This is in line with the non-exhaustive list of subject areas in the ICLR Call for Papers at https://iclr.cc/Conferences/2020/CallForPapers . This list includes:
> - unsupervised, semi-supervised, and supervised representation learning
> - visualization or interpretation of learned representations
> - applications in vision, audio, speech, natural language processing, robotics,
> - neuroscience, computational biology, or any other field
>
> As a matter of reference, here is a small selection of comparable papers that were accepted at ICLR in previous years:
>
> https://openreview.net/forum?id=HJvvRoe0W
> The paper shows the benefits of a novel pre-processing technique for a new CNN architecture specializing in predicting the folding structure of DNA molecules.
>
> https://openreview.net/forum?id=ryl5khRcKm
> The authors present a tailor-made CNN architecture in the application domain of protein localization, which outperforms incumbent architectures as well as domain experts.
>
> https://openreview.net/forum?id=HJtEm4p6Z
> The presented model improves upon state-of-the-art text-to-speech systems by allowing more efficient training. The authors further tackle common problems of the synthesis task.
>
> Conformably, our paper validates some of the most recent explainability techniques for CNNs in the domain of behavioral biology and demonstrates their value. Furthermore, it details and justifies the approach taken, which can be very valuable for practitioners in other fields. For these reasons, we believe ICLR to be a suitable venue for the publication of our paper.

---

### Decision · Program_Chairs · 2019-12-19

**Decision:**

Accept (Poster)

**Comment:**

This paper presents a case study of training a video classifier and subsequently analyzing the features to reduce reliance on spurious artifacts. The supervised learning task is zebrafish bout classification which is relevant for biological experiments. The paper analyzed the image support for the learned neural net features using a previously developed technique called Deep Taylor Decomposition. This analysis showed that the CNNs when applied to the raw video were relying on artifacts of the data collection process, which spuriously increased classification accuracies by a "clever Hans" mechanism. By identifying and removing these artifacts, a retrained CNN classifier was able to outperform an older SVM classifier. More importantly, the analysis of the network features enabled the researchers to isolate which parts of the zebrafish motion were relevant for the classification.

The reviewers found the paper to be well-written and the experiments to be well-designed. The reviewers suggested a some changes to the phrasing in the document, which the authors adopted. In response to the reviewers, the authors also clarified their use of ImageNet for pre-training and examined alternative approaches for building saliency maps.

This paper should be published as the reviewers found the paper to be a good case study of how model interpretability can be useful in practice.